# Brain–Computer Interface (BCI) Control of a Virtual Assistant in a Smartphone to Manage Messaging Applications

**DOI:** 10.3390/s21113716

**Published:** 2021-05-26

**Authors:** Francisco Velasco-Álvarez, Álvaro Fernández-Rodríguez, Francisco-Javier Vizcaíno-Martín, Antonio Díaz-Estrella, Ricardo Ron-Angevin

**Affiliations:** Departamento de Tecnología Electrónica, Universidad de Málaga, 29071 Málaga, Spain; afernandezrguez@uma.es (Á.F.-R.); fjvizcaino@uma.es (F.-J.V.-M.); adiaz@uma.es (A.D.-E.); rron@uma.es (R.R.-A.)

**Keywords:** brain–computer interface (BCI), assistive technology, P300, messaging applications, WhatsApp, Telegram, e-mail, short message service (SMS)

## Abstract

Brain–computer interfaces (BCI) are a type of assistive technology that uses the brain signals of users to establish a communication and control channel between them and an external device. BCI systems may be a suitable tool to restore communication skills in severely motor-disabled patients, as BCI do not rely on muscular control. The loss of communication is one of the most negative consequences reported by such patients. This paper presents a BCI system focused on the control of four mainstream messaging applications running in a smartphone: WhatsApp, Telegram, e-mail and short message service (SMS). The control of the BCI is achieved through the well-known visual P300 row-column paradigm (RCP), allowing the user to select control commands as well as spelling characters. For the control of the smartphone, the system sends synthesized voice commands that are interpreted by a virtual assistant running in the smartphone. Four tasks related to the four mentioned messaging services were tested with 15 healthy volunteers, most of whom were able to accomplish the tasks, which included sending free text e-mails to an address proposed by the subjects themselves. The online performance results obtained, as well as the results of subjective questionnaires, support the viability of the proposed system.

## 1. Introduction

There are several diseases that cause severe impairment of motor skills in affected patients, such as amyotrophic lateral sclerosis (ALS) or chronic Guillan–Barré syndrome [1,2]. These limitations can lead to patients having difficulty in interacting with their environment. To solve this problem, assistive technology (AT) can be used, which can facilitate user interaction with the mentioned environment [3].

In general, AT can be used to control multiple devices, such as a wheelchair, a home automation system, or a verbal communication system [4]. A key point of AT is that it must be adapted to the abilities of the patient. Therefore, AT should be able to be controlled through those output channels that the patient still has preserved, such as the voice, the eye gaze, movements of a finger, the head, the cheek, or the tongue; these examples of AT systems rely on the use of input instruments such as microphones, eye-trackers, mechanical keyboards, head-pointing devices or low-pressure sensors, respectively [5]. A drawback to note is that, in severe and progressive motor limitations (as is the case in ALS), most of these examples of AT may no longer be useful because they depend on some type of muscular channel that may be affected in the patient, in either the initial or final stages of the disease.

Brain–computer interfaces (BCI) are a type of AT that uses the brain signal of users to establish a communication and control channel between them and an external device (usually a computer) [6,7]. Therefore, the main advantage of these systems is that a priori they do not require muscular control for their management. Thus, this technology may be a suitable option for those people who have completely lost the ability to move their muscles. The neuroimaging technique most used by BCIs is electroencephalography (EEG), possibly due to its relatively low cost and high temporal resolution [8]. Therefore, the present work will make use of the EEG as an input signal of a BCI system. There are several well-known EEG signals that are usually used to control BCIs, some of which will be mentioned next (see [9] for a recent review).

Regarding endogenous BCIs (systems based on the self-regulation of the EEG in the absence of any external stimuli), the most extended are those based on the variations produced in the sensorimotor rhythms (SMR) due to motor imagery (MI) tasks.

On the other hand, exogenous BCI (systems that use external stimuli for eliciting an EEG response) mainly rely on steady-state evoked potentials (SSEP) and on event-related potentials (ERP). SSEP present steady-state responses to periodic stimuli, such as the steady-state visual evoked potentials (SSVEP) and the auditory steady-state responses (ASSR). The ERP are changes in the EEG that are elicited as a response to a determined event. The most usual ERP used as input for BCIs is the visual P300 evoked potential, possibly due to its high performance, the number of output commands allowed, and the short time necessary for calibration [8]. Furthermore, in the specific field of BCI spellers (in which this proposal fits) the most used input is P300 [10], with a 65% of presence. Although other inputs as SSVEP offer similar or better performance, they present some drawbacks when it comes to paradigms with a high number of elements; according to [10], “a higher number of targets in SSVEP-based BCI increases the spelling speed but also increases eye fatigue and target misclassification”. In the present proposal there are menus with up to 49 elements (see Section 2.4). Other modalities of evoked potentials, such as ASSR, present a lower performance than visual BCIs due to the small amplitude and modulation of ASSR [11]. Thus, the P300 ERP was selected as the input signal to control the proposed paradigm. P300 is a positive potential generally located in the parieto-occipital areas that appears about 250–500 ms (although the range can vary depending on numerous factors) after the presence of both an expected stimulus and a rare one [12]. Usually, the P300 is evoked through an oddball paradigm, in which the available items are highlighted pseudo-randomly while the user pays attention only to the desired item, thus resulting in a P300 potential after the stimulation of this desired item. After a predetermined number of iterations, the system averages the resulting EEG and determines which item the user wanted to select. This concept was adopted by [13] to propose a paradigm to control a text speller.

Making the leap to useful and practical applications for the target population of BCI systems is an important objective. Numerous works have made efforts in this direction, focusing on the control of devices used daily by patients, such as a medical bed, a wheelchair, smart lights, a web browser, or messaging systems. Table 1 includes those proposals that, to our knowledge, have implemented a BCI system based on the EEG signal for the control of patients’ everyday devices (systems that presented simulated environments have not been included). Note that two of these systems [14,15] do not only use EEG as input but use electromyography (EMG) or electrooculography (EOG) as well, resulting in hybrid BCIs.

**Table 1 sensors-21-03716-t001:** EEG-based BCI proposals that include daily use systems.

Work	EEG Signal	Controlled Devices
[16]	P300	36 commands not specified
[17]	P300	TV, DVD player, hi-fi system, multimedia hard drive, lights, heater, fan, and phone
[18]	Conceptual imagery	Kettle, shutters, TV, and light
[19]	P300	lights, doors, fan, camera, media player, and predefined websites
[20]	Auditory ERP	TV, air conditioner, and emergency call
[21]	ASSR	Smart bulb and fan
[22]	P300	Twitter and Telegram
[23]	SSVEP	Robotic vacuum, air cleaner, and humidifier
[24]	SMR	Medical call, service call, catering ordering, TV, and two air conditioners (wall-hanging and cabinet)
[25]	Visual ERP	TV, air conditioner, and emergency call
[26]	P300	TV
[14]	SSVEP + temporalis muscle (EMG)	Wheelchair, nursing bed, TV, telephone, curtains, and lights
[15]	SMR + blinks (EOG)	Speller, web browser, e-mail client, and file explorer

The particularity of these studies lies in establishing a communication channel between the BCI application (e.g., BCI2000 [27]) and the software that controls the device that is intended to be used. Commercial or everyday devices are not adapted to be controlled directly through a BCI system. Three of the mentioned papers implemented an interaction with a smartphone [20,22,25], and the three used Bluetooth to send control commands from the BCI to the smartphone. In [20,25] authors connected an Arduino Bluetooth module to a computer to send the commands and installed a custom-made application in the smartphone to receive and interpret these commands. In the case of [22], a computer directly sent the control commands via Bluetooth to the smartphone, but unfortunately, the authors did not provide the details on how the smartphone received and interpreted these commands (they only mentioned “The selected commands are sent in real-time to the final Android device via Bluetooth, which interprets them and provides visual feedback to the user”); we guess that they also used an application running in the smartphone to receive and interpret the control commands. Thus, in the case of sending control commands to several applications, a control module should specifically be designed to operate each application through its specific application programming interface (API), which is a drawback as it can limit the number of potential applications that can be used. A possible solution to this relies on the use of a virtual assistant, which would allow a communication channel with a smartphone that can be accomplished more easily and reaching many more devices. Nowadays, smartphones can be controlled through virtual assistants such as Google Assistant, Alexa, or Siri [28], which allow the generic control of numerous applications and external devices. Unlike the previous case of sending control commands in a programmed way, with the use of virtual assistants it is not necessary to implement any specific API since the virtual assistant is ready to communicate with many applications and devices. All these virtual assistants have in common that they can be controlled through voice commands. Therefore, if the BCI system were able to send commands through voice, the virtual assistant could interpret them and execute the correct action on the device to which it is connected (e.g., a smartphone). There are some works that have already explored the idea of using voice commands sent to virtual assistants to facilitate integration between applications. Outside the field of BCI, the work of [29] can be highlighted. These authors used a proximity sensor on the fingers, feet, or head (depending on the patient) to select commands in an application with a graphical interface—called MacroDroid—which later allowed text to be converted to speech to verbalize the users’ selections. The voice assistant used for this work was Google Assistant. Thanks to this system, patients with motor impairment were able to control WhatsApp and YouTube applications. This work was based on a reduced set of possible actions: for WhatsApp, three contacts to choose and three predetermined possible messages to send; for YouTube three possible music/videos alternatives and four alternative-related events. In relation to the works that have used a BCI, the work of [21] has been the only one, to our knowledge, that has used a voice assistant (Alexa) to control two devices (a light bulb and a fan) through a signal obtained from the EEG of the user without involving muscle control. As stated by the authors: “voice commands, rather than electronic activation, allow simple integration with current existing devices on smart hubs such as Google Home and Amazon Alexa”. In the present work, the voice assistant used is Google Assistant.

The loss of communication (mainly with family and caregivers) is considered by ALS patients as even more negative than the loss of physical aspects [30,31]. Therefore, this work will focus on the use of a BCI that could allow patient communication through some of the most common messaging systems on a smartphone: WhatsApp, Telegram, e-mail, and short message service (SMS). According to data reported by the companies themselves, in February 2020 WhatsApp reached “more than two billion users around the world” [32], while Telegram reported 400 million users in April 2020 [33]. Regarding e-mail, in 2020 there were over 4 billion e-mail users worldwide [34]. Finally, as the SMS service is compatible with all smartphones and offered by all operators, it has 3.8 billion potential users in 2021 [35].

To configure the BCI, the UMA-BCI Speller tool was used, an application that allows easy control of the visual parameters of the interface [36]. Thanks to this tool (which includes a text-to-speech function), all the necessary options could be added to control the different messaging services through the voice-controlled Google Assistant. The use of a versatile and easily configurable tool, which allows the addition or removal of possible devices to control, is a key aspect of this platform since the variety of devices that can be controlled will be limited by the voice assistant used.

In short, the objective of this work is to create a communication bridge between the UMA-BCI Speller platform and the messaging services of WhatsApp, Telegram, e-mail, and SMS, through the use of Google Assistant on a smartphone. Users could send free-text messages by spelling them (i.e., without being limited to a reduced set of possible messages, unlike proposals such as [29]). To our knowledge, this is the first BCI proposal that includes the use of WhatsApp and SMS. Furthermore, throughout this work the ability of the virtual assistant to interpret the commands issued will be evaluated, thus allowing the detection of possible errors. We will also study how these errors could be solved for the implementation of future applications.

## 2. Materials and Methods

### 2.1. Participants

The study initially involved 15 healthy voluntary participants (S1–S15), but three of them were dropped for different reasons: (i) one subject had needle panic (S3), so impedance adjustment could not be performed and thus the quality of the EEG was very poor, (ii) two participants lost control during the experiment and achieved an accuracy below 70% (S11 and S15), which is usually considered the minimum necessary to enable an efficient communication system [37]. Thus, the results of 12 subjects (aged 21.6 ± 2.5 years, 6 females, all of them students at the University of Málaga) were finally analyzed in the Section 3. The study was approved by the Ethics Committee of the University of Malaga and met the ethical standards of the Helsinki Declaration. According to self-reports, none of the participants had any history of neurological or psychiatric illness. Participants received a monetary remuneration of EUR 5 and all of them provided written consent.

### 2.2. Data Acquisition and Signal Processing

The EEG was recorded at a sample rate of 250 Hz using the electrode positions: Fz, Cz, Pz, Oz, P3, P4, PO7, and PO8, according to the 10/20 international system. All channels were referenced to the left mastoid and grounded to position AFz. Signals were amplified by an acti-CHamp amplifier (Brain Products GmbH, Munich, Germany). All aspects of the EEG data collection and processing were controlled by the BCI2000 system [27]. No artifact detection or correction techniques were applied.

The visual P300 evoked potential was the EEG signal used to control the BCI. P300 paradigms do not need any user training, but calibration is required in order to obtain the subject-dependent parameters for the online experimental part. Thus, a particular analysis of every subject’s EEG was performed in order to carry out the feature extraction and to obtain a subject-dependent classifier. This analysis consisted of a stepwise linear discriminant analysis (SWLDA) of the calibration data through the BCI2000 tool named P300Classifier [38,39]. In this case, features are signals at a particular location (EEG channel) and time after the stimulus. A detailed explanation of the SWLDA algorithm can be found in the P300Classifier user reference [39], where it is summarized as a process “to obtain a final linear model that approximately fits a set of data (stimulus) by using multiple linear regressions and iterative statistical procedures, thus selecting only significant variables that are included in the final regression”. The default configuration was used (60 for the maximum number of features, and 0.1 and 0.15 as maximum *p*-value for the respective inclusion or exclusion of a feature in the model). The time interval analyzed was also a default value of 0–800 ms after the stimulus presentation. As a result of this analysis, the subject-dependent weights of a classifier were obtained; this classifier was applied to the EEG in order to determine which item the subjects attended to.

In order to minimize the effect of the noise of the EEG, the EEG source was band pass filtered in the band 0.1–9 Hz (a first-order infinite impulse response (IIR) for the high pass filter and a second-order Butterworth IIR for the low pass filter) and a notch filter was set at 50 Hz (two third-order Chebyshev filters).

It is worth mentioning that other decoding systems based on machine learning, such as deep learning, have been recently applied to BCI systems with promising results (see the review in [40] for details). However, the objective of this proposal is not related to signal processing algorithms, but to present a new modular architecture using the virtual assistant of a smartphone to control some common applications; for this reason, we used the standardized SWLDA from BCI2000 as a classifier.

### 2.3. System Implementation

The aim of the BCI system was to generate voice commands that could be interpreted by a virtual assistant running in a smartphone. These voice commands were intended to read and send messages through various messaging services. In order to achieve that, a BCI system was implemented that could generate these commands in text form and convert them into voice. On the one hand, a laptop ran the software that presented the stimuli and registered and analyzed the EEG. This software was the UMA-BCI Speller, a free tool that wraps BCI2000 and simplifies its configuration and use. This software was used to spell and convert the control commands to voice. On the other hand, a virtual assistant was running on a smartphone. This virtual assistant was Google Assistant and it received and interpreted the voice commands, performing the corresponding action. The system implementation is shown in Figure 1.

The UMA-BCI Speller includes a text prediction function that may help users when spelling words. As users choose the characters of a word (starting with the first character), the system proposes several predicted words based on the characters already written and the probability of occurrence based on a Spanish language specific corpus (easily generalizable to other languages, e.g., an English language corpus can be downloaded from [41]).

Once users had completed the spelling of the text command to send, they had to select a confirmation item in the interface so that the system could convert this command into speech. The Windows 10 Narrator (a text-to-speech feature) was used, particularly the voice named “Microsoft Helena” from the Spanish voice catalog.

To avoid the influence of ambient noise on the understanding of the command by the virtual assistant, the voice commands were sent to the smartphone via a cable connection (using a mini-jack audio cable) connecting the laptop audio output with the smartphone microphone input. The output volume of the laptop was fixed throughout the whole experiment, so the assistant always received the same level of audio. Since the voice commands were generated using a synthesized voice, the assistant was expected to interpret the same received commands in the same way. However, as we will detail in the specific Section 3.3. this was not always the case.

As the virtual assistant used in the experiment was Google Assistant, each command started with the words “Ok Google…”, which is one of the wake-up keywords of the assistant. Two main types of voice commands were used: (i) commands asking the assistant to read the received messages, e.g., “Ok Google, read my WhatsApp messages”, and (ii) commands asking the assistant to send a message using one of the messaging services installed in the smartphone, e.g., “Ok Google, send a Telegram to Francisco hello”. In addition to these, other voice commands were used that were needed to confirm or cancel actions, as we will explain in the next section.

### 2.4. Control Paradigm

In order to send a control command to the virtual assistant, users had to select items from different menus. The selection of an item followed the usual procedure in a P300 row-column paradigm (RCP): users had to pay attention to the desired item (within a matrix of possible items) and mentally count the number of times it was highlighted. The duration of the stimuli was 192 ms and the stimulus onset asynchrony (SOA) was 224 ms. The timing of each selection for all the menus was the same, as all the interfaces consisted of a 7 × 7 matrix, even though in three of them there were dummy items (items that had no effect when selected). An item was selected after all the seven rows and columns were highlighted a certain number of times (explained in Section 2.5).

Several menus were implemented that allowed subjects to gradually form a sentence that would finally be converted to speech by the Windows 10 Narrator voice synthesizer (from now on, this conversion will be denoted as “speak”). The sentence to be spoken was present in the interface so that subjects decided when to indicate to the system to speak it. Some items added several predetermined words to this sentence, while other items were present to add individual letters to spell a word.

As mentioned above, for the stimuli presentation and menu navigation the UMA-BCI Speller was used. Four different interfaces were implemented, as shown in Figure 2.


(a)No Control (NC) menu (Figure 2a). This was a 7 × 7 matrix in which only one item was a valid command, and the other 48 items (“X”) were dummy commands. The objective of this menu was to allow subjects to remain in a state where they could rest without generating control commands; the term “no control” is generally used in asynchronous systems to refer to such a state. The only valid command was “IC” (in the center of the interface), whose selection changed the menu to an Intentional Control menu. The probability of unintentionally selecting this item and thus changing the menu was 1/49 ≅ 2%. No voice command was generated in this interface.(b)Intentional Control (IC) menu (Figure 2b). This was the main menu of the system, where subjects could choose what action they wanted to select. In a 7 × 7 matrix, ten valid options were available (the remaining 39 options were dummy non-visible items). These ten options can be grouped into three categories:Send messages. This group consisted of four commands: “Send WA”, “Send TG”, “Send SMS” and “Send Mail”, that enabled users to send a message using WhatsApp, Telegram, SMS, or e-mail, respectively. Once one of these commands was selected, the system wrote part of the sentence to be spoken, “*Ok Google, send a WhatsApp to*”, “*Ok Google, send a Telegram to*”, “*Ok Google, send an SMS to*” or “*Ok Google, send an e-mail to*” and then changed to a Spelling menu (Figure 2c) so that the user could next spell out the receiver of the message and the message itself. Please note that throughout the manuscript we use italics to indicate the spoken commands sent to the assistant, as well as the assistant’s responses.Read messages. Three commands formed this group: “Read WA”, “Read TG” and “Read SMS” used to read the messages received through WhatsApp, Telegram, and SMS, respectively. The selection of one of these commands made the system speak the corresponding sentence: “*Ok Google, read my WhatsApp messages*”, “*Ok Google, read my Telegram messages*” or “*Ok Google, read my SMS messages*”. After the sentence was spoken, the system deleted it and automatically switched to the NC menu so that subjects could listen to the received messages, if any. After the virtual assistant read each received message, it asked the users if they wanted to reply to it or not. To do this, subjects first had to change to the IC menu where two commands were available related to replying to messages. These commands will be explained in the third group. After canceling or replying to each message, the system continued to read the remaining messages, if any.Other commands. In this group, three commands were included: “Reply”, “Cancel”, and “NC”. The “Reply” command allowed users to reply to a WhatsApp, Telegram or SMS received message after the system read them. Once this command was selected, the system wrote the sentence “*Ok Google, reply*” and switched to the Spelling menu so that users could complete the sentence with the desired response (in a similar way to what was done with the “Send” commands); please note that in this case, it is not necessary to specify the receiver of the message to be sent. The “Cancel” command was used to indicate to the system that the user did not want to reply to a received message. Once it was selected, the system wrote and spoke the sentence “*Ok Google, cancel*” and then deleted it and switched to the NC menu. Finally, the “No Control” command was presented in order to allow subjects to voluntarily change to the NC menu, in case they wanted to take a rest.(c)Spelling menu (Figure 2c). When users selected one of the “Send” or “Reply” options from the IC menu, the system changed to the Spelling menu (after adding some predetermined text). Here, the users could spell out the receiver and the message to send (or just the message in the case of the option “Reply”). It is worth mentioning that the message had to be spelled right after the receiver, only separated by a space. This menu consisted of a 7 × 7 matrix with spelling and control commands. The first six columns and rows corresponded to specific characters to be added (English alphabet letters and numbers). The last column was used to provide subjects with seven predicted words (this column used a different text color, as shown in Figure 2c). The last row contained two characters (“SPC” (space) and “,”), two delete commands (“Del.” to delete a single character and “Del. W” to delete a complete word), and two control commands (“OK” and “IC”). The command “OK” was used to indicate to the system that the receiver (if needed) and the message to send were complete so that the written sentence could be spoken and interpreted by the virtual assistant. The “IC” command was used to return to the IC menu without generating any voice command (this was useful if a subject entered this menu unintentionally). The selection of “OK” or “IC” caused the currently written sentence to be deleted (after speaking it in the case of “OK”), so a confirmation menu was offered to subjects in order to avoid undesired selections of these two commands.(d)Confirmation menu (Figure 2d). Two valid commands were available (among 47 non-visible dummy options) in a 7 × 7 matrix, “Confirm” and “Back”. On the one hand, the “Confirm” command was used to corroborate the previous selection in the Spelling menu (that is, “OK” or “IC”). In the case of confirming an “OK” command, the system spoke (and deleted) the complete sentence so it could be interpreted by the virtual assistant, and it changed to the NC menu. In the case of confirming an “IC” command, the system deleted the written sentence and switched to the IC menu. On the other hand, the “Back” command was used to return to the Spelling menu in order to continue writing the sentence to be sent to the virtual assistant.


### 2.5. Procedure

Subjects who joined the experiment received a video (10 min long) with practical details about the experimental process (regarding the EEG acquisition, the P300 potential paradigm, the navigation menus, and the tasks to be completed). The experiment had a duration of approximately 90 min and consisted of three parts: (i) a calibration session used to obtain the subject-dependent parameters of the EEG classifier, (ii) an online experimental part, and (iii) a final part in which the subjects filled out three questionnaires related to their subjective experience controlling the interface.

The calibration consisted of paying attention, without feedback, to 12 predetermined items (“DOMOTICA2021”) in the Spelling menu (Figure 2c). In this part, the number of sequences (i.e., the number of times that each row and column were highlighted) was fixed at eight, so each item was highlighted 16 times. The calibration session lasted around 6 min. After the calibration, an SWLDA analysis was performed to obtain the subject-dependent P300 classifier. According to the accuracy results obtained with such a classifier, the number of sequences was also adjusted for each subject. The criterion to choose this number of sequences was to maximize the written symbol rate (WSR), as proposed in [42,43]. However, this criterion was adapted so that the minimum number of sequences was three.

The online experimental part consisted of four tasks to be performed using different messaging services. In order to assess the subjects’ control, they started in the NC menu and they were asked to remain in this state for one minute before performing the first task. Once the four tasks were completed, they were asked to remain again for one minute in the NC state. 

In order to perform each messaging task, the users needed first to change to the IC menu before selecting the corresponding command. After the completion of each task, the system returned to the NC menu, where the subjects were instructed to remain as long as they considered it necessary to rest before undertaking the next task. The tasks to perform were written down on a paper next to the subjects so that they could take a look if they forgot the details. The messaging tasks were related to contacts already present in the used smartphone agenda. The four specific tasks were carried out in the same order and are listed next. The specific commands required to complete each task are detailed in Table 2.

To send a WhatsApp message to a contact named “Francisco” with the content in the Spanish language “experimento en la universidad” (in English, “experiment at the university”). This task was the same for all subjects. The receiver’s name as well as the words “experimento” and “universidad” were proposed as predictions by the system when one, three, and two characters were selected, respectively. A minimum of 19 actions were needed to complete this task.To check the received SMS messages and reply to any message coming from a contact named “*Ricardo*”. This was the shortest task, as no SMS were received during the experiment. Only two actions were needed to complete this task.To check the received Telegram messages and reply to any message coming from a contact named “*Álvaro*”. In this case, there was only one received message and its sender was Álvaro (this message was sent just before each experiment), so all subjects had to reply to it. The incoming message was a question: “*¿Cuál es tu comida favorita?*” (in English, “what is your favorite food?”). This task did not have a minimum number of actions to perform, as the answer to the question was different for every subject (they did not know the question in advance so they improvised the answer).To send an e-mail to a contact that the subjects decided with the content that they also decided. The receiver contact was previously registered in the smartphone agenda and subjects were asked to write down the chosen message before the experiment (in order to facilitate the evaluation). They were instructed to choose a 20- to 30-character message.

### 2.6. Evaluation

The present work included two types of metrics to evaluate the control of the system. On the one hand, the performance measures aimed to quantify the objective accomplishment of the tasks carried out by the user. On the other hand, the questionnaires were aimed to describe the subjective user experience of the control of the system.

#### 2.6.1. Performance

The performance of the system was assessed for both the calibration session and the online session. The performance of the calibration session was used to adapt the number of sequences according to the criteria presented in Section 2.5. However, it should be remembered that in the calibration session there was no feedback since the classifier parameters had not yet been calculated. The performance in the online session allowed us to know how the actual control of the system was. The objective measures used to assess the performance were: (i) the accuracy, for both sessions, (ii) the information transfer rate (ITR) for the online session, (iii) the WSR for the calibration session, (iv) the output characters per minute (OCM) for the online session, and (v) also for the online session, the absolute values of specific variables such as the number of commands selected or the time to complete each task.

Next, the general measures of system performance—accuracy, ITR, WSR, and OCM—will be described. First, accuracy is defined as the number of correct selections divided by the total number of selections and is usually expressed as a percentage. Second, the ITR is the number of bits transmitted per second and provides a more general evaluation than accuracy since, besides considering it, it takes into account the number of elements available in the interface as well as the time needed for each selection, which depends on the number of sequences used [44]. Specifically, the ITR is calculated following the equations:$$B={log}_{2}N+P{log}_{2}P+\left(1-P\right){log}_{2}\frac{1-P}{N-1}$$
$$ITR=\frac{B}{T}$$
where *B* is the number of bits transmitted per selection, *P* is the accuracy, *N* is the number of elements available in the interface (49 elements, including dummies, since all menus consisted of a 7 × 7 matrix) and T is the time required (minutes) for each selection. Third, the WSR is a variation of the ITR that assumes that the user has to correct errors in the selection of commands, and is therefore defined as a more ecologically valid measure [42]. This measure has been widely used by other proposals on BCI systems in the calibration session to select the optimal number of sequences in the subsequent online session (e.g., [45,46,47]). The WSR was calculated following the equations:$$SR=\frac{B}{{log}_{2}N}$$
$$WSR=\left\{\begin{array}{c} \frac{2SR-1}{T},\;SR>0.5\\ \;\;0,\;SR\leq 0.5 \end{array}\right.$$
where *SR* is the symbol rate, and the other elements have the same meaning as for the equations related to the ITR. Finally, the OCM was introduced for the first time by [48] and was calculated by dividing the number of written characters (including spaces) by the time required (minutes) to complete the tasks. Unlike [48], in this paper, this measure did not include the validation commands. This value was only calculated considering the selections made in the Spelling menu (if the participant left the menu prematurely, either by selecting the “IC” or “OK” commands, it was considered as an error). Regarding the number of written characters, this measure did not include the predetermined text of the voice commands, just the spelled characters for the receiver and the messages to send (see Section 2.4). For example, for Task 1, when the users selected the “Send WA” command, the system wrote the sentence “*Ok Google, send a WhatsApp to*”; these 29 characters were not included in the OCM calculation, just the spelled part regarding “*Francisco experimento en la universidad*” (39 characters). This measure provides an easily interpretable value of the efficiency of the system, i.e., the number of characters per minute that can be produced to advance in a written text. Also, unlike other measures mentioned (accuracy, ITR or WSR), it allows the utility of the text predictor implemented in the system to be assessed.

#### 2.6.2. Questionnaires

In order to assess the subjective user experience, three questionnaires were used: (i) the System Usability Scale (SUS) [49], (ii) the Raw NASA-TLX [50], and (iii) an ad hoc questionnaire designed by the researchers to extract additional information. All questionnaires were presented in Spanish, the native language of the participants.

The SUS was used to perform a general evaluation of the usability of the BCI system employed. It consisted of 10 items to be evaluated according to a 5-point Likert scale from 1 (strongly disagree) to 5 (strongly agree). The overall usability score provided by this questionnaire ranges from 0 to 100 (see [49] for the details of scoring the SUS). According to [51], a score of 70 is suggested as the acceptable minimum. The items used in the present work were the following (the name of the variable has been added at the end of each item in parentheses):I believe that patients and caregivers could use this application frequently (frequently used).I found this application unnecessarily complex (complex).I thought the app was easy to use (ease of use).I think I would need help from a tech-skilled person to use this application (technical support required).I found the various functions in this system were well integrated (well-integrated).I think the application is very inconsistent when executing the various actions (inconsistent).I would imagine that most people would learn to use this system very quickly (easy to learn).I found the application very cumbersome to use (cumbersome).I felt confident using this application (confident).I needed to learn many things before I was able to use this application (knowledge required).

The Raw NASA-TLX measures the user workload produced by the control of the system. This questionnaire is a modification of the NASA-TLX [52]. This modification has been widely used and consists of shortening the test to reduce the time required [50]. The Raw NASA-TLX is a multidimensional questionnaire with six subscales that are scored from 0 to 100 (mental demand, physical demand, temporal demand, performance, effort, and frustration), in intervals of five units. Higher scores will mean a higher workload. The endpoints for each subscale are “very low/very high” except for the performance subscale, which has “perfect/failure” endpoints. The total workload was calculated by averaging the scores obtained in each of the subscales. The items used were the following (the name of the variable has been added at the end of each item in parentheses):How mentally demanding was the task? (mental demand)How physically demanding was the task? (physical demand)How hurried or rushed was the pace of the task? (temporal demand)How successful were you in accomplishing what you were asked to do? (performance)How hard did you have to work to accomplish your level of performance? (effort)How insecure, discouraged, irritated, stressed, and annoyed were you? (frustration)

In addition to the SUS and the Raw NASA-TLX, a small ad hoc questionnaire was prepared for an additional evaluation of aspects that were considered relevant. The answers given to this questionnaire could be used to extract useful information that could be considered in future proposals or modifications of the presented system. The questionnaire consisted of five open-response items. The following items were used (the name of the variable has been added at the end of each item in parentheses):List up to three negative features of the interface (negative features).List up to three positive features of the interface (positive features).Would you consider adding any other messaging application, and if so, which one? (other messaging application).Would you add any functionality to the system?If so, which one? (other functionality).Additional comments (additional comments).

## 3. Results

This section presents the results obtained from the performance metrics and the subjective questionnaires. Section 3.3 will describe some issues related to the virtual assistant that resulted in errors when handling the messaging applications.

### 3.1. Performance

#### 3.1.1. Calibration

Figure 3 illustrates the accuracy and WSR obtained by participants during the calibration task. On average, most of the participants (7 out of 12) reached the 70% criterion proposed by [37] with only two sequences. From sequence 5 onwards, all users were above this criterion. Using the maximum number of sequences allowed (i.e., 8 sequences), all participants except S1 reached 100% accuracy. Regarding the WSR, it can be observed that, on average, the maximum peak is obtained with 3 sequences (the minimum number of sequences that it was decided that could be used in the online session), when the average accuracy was equal to 86.83%. The results obtained in performance are consistent with those shown in Figure 4 for the grand average ERP waveform, where the signal associated with the target and non-target stimuli can be clearly discriminated.

#### 3.1.2. Online Session

The average number of sequences chosen to perform the online session was 4.25 ± 1.49. Table 3 shows the results obtained for each of the four tasks in terms of the number of selections made by the users (i.e., selected commands) and the time taken to complete them, as well as the overall accuracy. Each task that involved a spelling part (Task 1, Task 3, and Task 4) took in an average of 8–10 min and required approximately 23–30 selections. On the other hand, for Task 2, all participants except one required less than one minute (S7 needed 95 s) and a maximum of 3 selections. The overall accuracy and ITR obtained in these four tasks were 86.14% and 21.69 ± 6.73 bits/min, respectively.

Regarding the task of waiting for one minute in the NC menu, both initial and final, all participants accomplished the required time except for participant S8 in the initial task. Therefore, it can be stated that the proposed NC state was 96% effective (only one error out of 24 times tested, two times for each participant), which is close to the result expected, since the threshold of chance for an FP for the “IC” command is approximately 2% (one out of 49).

In reference to the phases in which the user could rest, they remained in the NC menu for a total time—adding up the time before starting Task 2, Task 3, and Task 4—equal to 63.34 ± 41.37 s (which corresponds to an average of 3.3 ± 1.7 random selections of dummy commands; see Figure 2a). The commands to delete a letter (“Del.”) or a word (“Del. W”) were selected throughout the experiment an average of 6.5 ± 4.5 and 0.9 ± 1.2 times, respectively. Likewise, the text predictor was used an average of 9.17 ± 1.4 times by each user. The number of total incorrect selections was 12.67 ± 10.08, which included 3.42 ± 3.32 selections of dummy elements.

The following two measures will only consider the results in the spelling parts, i.e., related to Task 1, Task 3, and Task 4. On the one hand, the average number of commands selected to complete all these parts was 65.33 ± 12.62; on the other hand, the number of symbols written to complete them was 83.25 ± 5.75. Therefore, it is shown that the prediction system assists users, as more characters were written than commands were selected. The average total time to complete the spelling parts was 1282.89 ± 353.75 s. Concerning the OCM, the average value of the individual OCMs obtained by the participants was 4.11 ± 0.94.

### 3.2. Questionnaires

All the subjects included in the study answered the three proposed questionnaires after completing the experimental tasks.

#### 3.2.1. System Usability Scale

Regarding the subdimensions assessed with the SUS, all the positive items (odd items: frequently used, easy to use, well-integrated, easy to learn, and confident) of this questionnaire were scored above 4 out of 5, while the negative items (even items: complex, technical support required, inconsistent, cumbersome and knowledge required) did not obtain average values above 2 out of 5 (Figure 5). Likewise, the average score for the overall usability was 82.5 ± 15.63 (Figure 6). Only 2 of the 12 participants (S4 and S5) presented a score below 70. Therefore, it can be concluded that the system offered a highly adequate/positive rating by the users.

#### 3.2.2. Raw NASA-TLX

The average scores for each of the subdimensions of the Raw NASA-TLX questionnaire can be seen in Figure 7, where the two variables that contributed most to the user’s workload were effort and temporal demand. This could imply that the task may have been too fast-paced and that the participants may have been struggling to perform the task and obtain the highest performance. The total workload, shown in Figure 8, obtained after averaging each of the subdimensions, was 31.55 ± 17.44; none of the participants had a score above 50, and two of them even scored below 10 (S6 and S10).

#### 3.2.3. Ad Hoc

For this questionnaire, the answers of the participants to each of the items will be described. Not all their answers have been reported, but only those that were considered relevant by the researchers.

(A)Negative features

Four participants questioned the visual aspect of the interface, arguing that it was not very attractive (S1), that the layout of the letters could be similar to a conventional keyboard (e.g., QWERTY layout) (S2), commented on the general layout of the elements in the interface (S13), or that the illuminations could blind light-sensitive people (S14). Participant S7 complained about the need to use a cap, which could be a problem for certain patients (e.g., use of assisted ventilation systems), as well as the discomfort of wearing it and the use of electrolyte gel. This same participant also criticized the portability of the system and said that it also required too much concentration for adequate control. Finally, two participants (S5 and S12) stated that the system was too slow.

(B)Positive features

The quality of the interface most valued by the participants was its ease of use and how intuitive it was (S1, S5–S10, S13, and S14). Also, two participants (S1 and S2) stated that the system was fast to complete the different tasks. Four participants (S1, S6, S8, and S10) appreciated a large number of options allowed and the flexibility of the system. Three participants (S6, S13, and S14) had positive comments regarding the ease of fixing mistakes thanks to the correction commands (delete letter or delete word) and the presence of confirmation menus. The use of the text predictor was positively rated by one participant (S9).

(C)Other messaging applications

Not many participants suggested other messaging applications and only Line (S8) and Facebook messenger (S12) were suggested. Also, participant S1 suggested the possibility of being able to write public messages on the Facebook and Twitter platforms.

(D)Other functionality

Two participants suggested that the dictionary should have a larger number of words (S1 and S12), and one of them also suggested that it could be adapted to the particular user (S1). Participant S12 asked that the application should allow the use of uppercase and lowercase letters, acute accents (which are quite common in Spanish and may change the meaning of words), and more punctuation marks. Also, participant S4 suggested the possibility of adding a pause between each menu change. In addition to suggestions related to the messaging system, one participant (S6) suggested the possibility of being able to add alarms or reminders in the application. Also, another participant (S1) suggested adding a counter to know when the flashes started.

(E)Additional comments

Only four participants made additional comments on the application. Participants S5 and S6 found the application to be a very useful idea, although S5 pointed out that some improvements would still be necessary for its practical use. Participant S12 specified that the controlled application could be useful for people with motor speech problems. He/she also added that, with proper implementation and increased speed, these systems could be the future of how to use a lot of electronic devices. Participant S8 stated that he/she was pleasantly surprised by the use of a voice assistant, and how easy and simple it was to control. Finally, participant S11 stated that the experiment was a nice experience.

### 3.3. Issues with the Virtual Assistant

Throughout the experiment, some problems related to the virtual assistant’s speech recognition arose. Even when the voice commands were generated with a synthesized voice in the laptop, using the same output volume level and with the audio signal fed to the smartphone through a wired connection, there were times when the same command was not interpreted in the same way. Next, Table 4 summarises these errors and afterward gives a complete description and the way these errors were fixed. A possible explanation for the errors can be found in the way the virtual assistant voice recognition works, which will be detailed in the Section 4.

Type 1 error. Regarding Task 3, we found a timing problem during the preliminary tests. After the user read the Telegram message with the question “*What is your favourite food?*”, the assistant asked, “*Do you want to reply?*”. After this moment, the user had five minutes to send the command “*Ok Google, reply* [message]”. After five minutes, the assistant no longer recognized the context of the command (a previous message to be replied to), so the response was not sent. As five minutes was too short to form the response using the BCI system, we implemented a simple routine in the assistant. A routine consists of a series of actions that are grouped and executed when the user speaks a determined trigger command. In this case, the routine (named “Wait a while”) was triggered when the user selected the “Reply” command in the IC menu. When this command was selected, the system spoke the sentence “*Ok Google, wait a while*”, which was the trigger for the mentioned routine. The routine consisted of reducing the smartphone volume to an inaudible level, asking the assistant to (silently) count to 100, then counting again to 100, then to 80 (which lasted almost five minutes), and then asking the assistant “*Can you repeat?*”. To this question, the assistant replied by repeating the question to the user “*Do you want to reply?*”, which gave the user another five minutes to reply. After spelling their response, when the users confirmed the sending of the message, the volume level of the smartphone was restored with a new command. The aforementioned “Wait a while” routine caused a new problem. If the user selected the “Reply” command by mistake, thus entering the routine, he/she could cancel the actions of the routine by selecting the “Cancel” command from the IC menu. This selection, in turn, triggered a second routine (named “Routine Cancel”) that stopped the five-minute count and restored the volume level. This new routine was triggered with the voice command “*Ok Google, routine cancel*” (in Spanish “*Ok Google, rutina cancelar*”). However, while the assistant was involved in the counting of the first routine, it seemed to have trouble understanding the voice trigger and interpreted “*routine ca*-” (missing the last two syllables in Spanish), while it could correctly interpret the same voice trigger in an idle state (not counting). The adopted solution was to include a second trigger in the same routine, so this second routine could be triggered with “*routine cancel*” or with “*routine ca*”.In relation to this time limit, an error occurred with subject S13 in this same Task 3. Instead of answering with a short message, this subject spelled the whole sentence “*la pasta es mi comida favorita*” (in English, “*pasta is my favorite food*”), so he/she took too long to form the sentence “*Ok Google, reply pasta is my favorite food*” and thus the assistant did not interpret it as a response to a Telegram message, but as new information to be saved. In consequence, the assistant replied “*Ok, I’ll remember that pasta is your favorite food*” instead of sending the Telegram response.Type 2 error. Task 4 required that each user proposed an e-mail address to be registered in the smartphone list of contacts. We decided to use the same name for the contact and to change the e-mail address with each subject so that all users had to spell the same receiver. However, the virtual assistant remembered the e-mail address of the previous subject, even when it had been changed in the contact list of the smartphone because in the Gmail servers this contact did not have the address properly updated. This caused subject S2 to send his/her e-mail message to an undesired address (the one chosen by S1). This error was fixed using different contact names for each subject.Type 3 error. As a consequence of the previous error, we decided to use the user identifier as the new contact name for each e-mail address in Task 4. We started naming these contacts as “Subject Three”, “Subject Four” and so on. However, with subject S4 the virtual assistant misinterpreted this two-word contact and it considered the word “Subject” as the receiver and the second word (“Four”, the specific number identifying this subject) as part of the message. Since two names starting with “Subject” (“Subject three” and “Subject four”) were registered in the contact list, the virtual assistant asked the user to specify, “*Which Subject do you want to send an e-mail to?*”. As the user could not answer this unexpected question, this e-mail was not sent. From subject S5 on, we corrected this error using a single-word name for each contact, the user’s first name. Although the naming for each contact using a single-word could be a disadvantage in real world scenario (for example, to differentiate between two people with the same name), this was chosen to standardize the spelling task for the users and to facilitate the voice recognition task by the virtual assistant.Type 4 error. The fix to the previous error worked correctly for two subjects, but subject S7′s given name was already registered in the smartphone. For this subject we registered as the contact name the user’s given name, to which we added the first syllable of the user’s family name, thus resulting in an unrecognizable word that the assistant did not interpret correctly. For this reason, subject S7′s e-mail was not sent. This error was fixed using an easily recognizable single-word name for each contact, the users’ family name.Type 5 error. Regarding Task 3, the voice command used to read the received messages was, in Spanish “*lee mis Telegram*” (in English, “*read my Telegrams*”). It was used for the participants S1 to S7; however, with S8 the virtual assistant started to misunderstand the command, so it was interpreted as “*lee mistela*” (in English, “*read mistela*”, a meaningless phrase since “mistela” is a kind of fortified wine), which sounds somewhat similar to “*lee mis telegrams*”. The consequence was that the assistant did not read the Telegram messages. In order to fix this problem, this control command (as well as the other “read” commands) was modified to be more explicit: “*lee mis mensajes de Telegram*” (in English, “*read my Telegram messages*”).Type 6 error. In Task 4, the voice command that was used to send an e-mail was of the type “*Ok Google, send an e-mail to* [receiver], [message]”, without specifying any e-mail subject. In the case of user S8, however, the virtual assistant interpreted that the spelled message was the subject of the e-mail and that the body of the e-mail was the voice command “*Ok Google, send it*”, which was used to confirm the sending of the e-mail. As a consequence, the e-mail was not sent.Type 7 error. Subject S11′s intended response to Task 3 was “*arroz*” (“*rice*”, in English); however, this subject made a mistake and sent the word “*arro*”, missing the last character. As “*arro*” is not a word in Spanish, the assistant corrected the mistake and interpreted the answer as “arroz”. Although this automatic correction cannot be exactly considered an error, we included it in this category as it corresponds to a difference between what the user spelled and what the assistant interpreted.Type 8 error. Subject S14′s spelled e-mail in Task 4 was “*hola Paula feliz Navidad*” (in English, “hello Paula merry Christmas”) but the assistant missed the first word and sent the e-mail without it.

## 4. Discussion

This section discusses and contextualizes with related literature the previously presented results regarding performance metrics, subjective questionnaires, and issues with the virtual assistant. It should be considered that these comparisons between studies have a rough approach, since the differences between studies are usually notable (e.g., EEG instrumentation used, duration of the task, the specific BCI paradigm, or the number of possible choices) and the relevant statistical analyses (e.g., significance tests) have not been performed.

### 4.1. Performance

The presented system is based on the well-known visual P300 RCP paradigm using flashing letters as stimuli, so the results, as expected, are close to those obtained in similar experiments. Here we reported an overall online accuracy of 86.14% and the correspondent obtained ITR was 21.69 bits/min. As a comparative example, one of our recent experiments [53] obtained an online accuracy of 82.74% and ITR of 22.46 bits/min also in an RCP paradigm based on flashing letters. Therefore, we could conclude that the context of messaging applications in which the RCP paradigm has been tested in this experiment did not affect the selection accuracy in comparison with a letter spelling context.

From the mentioned papers studying daily use systems in Table 1, the work of [22] is the most similar to the experiment reported here, as it also presented a system based on a visual P300 RCP to control messaging applications in a smartphone (Telegram and Twitter). They reported a mean accuracy of 92.3% and an OCM equal to 2.06 for the non-motor disabled subjects. The OCM reported here is almost double, 4.11, but it is important to remember that this value considers the total written characters, thus using the available prediction function, which is not present in [22]. Therefore, in order to establish efficient writing, this may indicate the importance of the text predictor, which should be employed by those systems that aim to serve as useful tools in real environments. A more suitable comparison could be established without considering the predicted characters, that is, by dividing the total number of selections by the duration of the tasks. This value has been reported in Section 3.2.1 and it corresponds to an average of 65.33 selections (in the spelling menus) in an average time of 1282.89 s, that is, 3.05 selections/min.

The results in Table 3 show that the subjects needed an average of 82.92 commands to complete the required tasks. A calculation of the minimum actions required to accomplish the tasks (without making any mistake) resulted in 61.17 actions, meaning that subjects used 35.5% additional commands compared to an error-free manual performance.

However, the objective of this work was not to offer a system with a high ITR or OCM but to show how the use of already tested tools might be combined to implement a BCI system capable of controlling the messaging applications of a smartphone. This objective was accomplished by 12 out of 14 participants (without considering the discarded subject with needle panic). It can be considered that in a real environment (i.e., not experimental) some factors can deteriorate the performance of the BCI system (e.g., attentional distractions or the absence of specialized technical staff). However, if the real environment is considered as one in which the user intends to use the system frequently, it should be noted that performance in a visual ERP-BCI can be improved by training [54]. Therefore, it would be convenient that future studies address this situation on prolonged use of these systems in real environments. Additionally, it should also be considered that in a real implementation of this system it is not convenient to apply the approach of automatically discarding those users who do not perform well. Each case should be studied specifically to find out why this inadequate performance is present and solve it.

It is possible that the results shown in performance may have been affected by fatigue. This effect has been previously reported in BCI studies based on the use of P300 [55]. However, in the present work, this effect could not be controlled since the tasks were different and the order was not counterbalanced between participants. That is, it cannot be known whether the initial tasks are better performed because of the absence of fatigue or because they were easier. Therefore, it would also be interesting for future work to address this possible problem of fatigue on performance with the aim of reducing or delaying it.

### 4.2. Questionnaires

From the proposals mentioned in Table 1, only those related to [14,15,18,22], have employed subjective measures for the assessment of the application used. However, none of these proposals employed the SUS; only [14,15] employed the NASA-TLX, and [18,22] used some ad hoc items. Subjective questionnaires may be susceptible to the presence of certain biases, such as social-desirability bias [56]. However, there is no particular reason to suspect that the present proposal is more affected by these effects than other works. Therefore, as far as possible, the results obtained in the present study will be contextualized with those obtained by the aforementioned studies.

The SUS scores reported for the application used were fairly positive. In the overall usability measure, a score of 82.5 ± 15.63 points was obtained, 70 being the threshold to establish a good level of usability according to [51]. Only two of the 12 participants had a score below the mentioned threshold. Likewise, the individual scores for each subdimension were also reasonably successful. All negative subdimensions scored between 1 and 2 points, and positive subdimensions between 4 and 5 points. From the scores obtained, it can be concluded that the application presented was easy to use, intuitive, and required minimal knowledge to control. Therefore, regarding these parameters, the application seems to obtain a rating similar to that achieved by [18] (ease of use and ease of learning, both above 5 out of 7) and [22] (intuitive and easy to understand, 4.79 out of 7, and not needing a manual, 4.93 out of 7) using ad hoc items.

The Raw NASA-TLX questionnaire, like the SUS, also offered a positive assessment of the application. The average total workload score was 31.55 ± 17.44 points, which can be considered reasonably low. In relation to the works that used this questionnaire, both [14,15] only reported the values related to the subdimensions and not the total workload. On the one hand, in [14] it was found that the subdimension that contributed most to workload was performance, with a score of around 50 and 80 points, while the remaining subdimensions were around 20 and 40 points on average. On the other hand, [15] reported that all the subdimensions scored between 20 and 40 points on average. The results offered by the present study were in approximately the same range, since all the average measures were also between 20 and 40 points, except for physical demand, which was slightly below (17.5 points). Therefore, the BCI application proposed in the present work for the control of external applications (messaging systems) requires a similar, if not lower, workload than that presented by previous studies.

Finally, although it was not quantitative data, the items related to the ad hoc questionnaire have provided certain indications of the strengths and weaknesses of the application. For example, according to the participants, it was clear that the interface should be improved aesthetically. However, it is also clear that the goal of making the application easy and intuitive to operate has been achieved. In addition, it seems that the proposed messaging applications were appropriate, as only two participants suggested other messaging applications (Line and Facebook messenger). Also, although outside the scope, the idea of extending the messaging system to the control of social networks (e.g., Facebook or Twitter), as suggested by one participant, is interesting. Regarding other functionalities, it would be appropriate to enlarge the dictionary corpus and improve the prediction system to adapt it to the specific user who handles it.

In short, the set of subjective questionnaires has provided valuable information, from which it can be concluded that the proposed system was easy to control and was a pleasant experience for most of the participants. However, the information obtained in this section can be used to improve the system further. Also, it would be appropriate to test the application presented in this work with patients with motor impairments to find out if it could be denoted as a useful tool for them as well.

### 4.3. Issues with the Virtual Assistant

In the Section 3.3 some issues related to the use of the virtual assistant were detailed. Mostly, they were misinterpretations of the spelled text. The explanation for that can be found in the way the virtual assistant’s voice recognition works. Virtual assistants use artificial intelligence (AI), machine learning, and natural language processing (NLP) to convert speech into text. NLP is based on deep learning (DL) and neural networks to recognize patterns, categorize, contextualize and translate them into text and perform a voice-based interaction [57]. DL application training and optimization rely strongly on stochastic procedures [58], so they can result in non-deterministic outcomes. This may be one reason that explains the fact that the same repeated input produces two different results. In the case of Google Assistant, the audio corresponding to the voice command is streamed to a server where it is transcribed by the server-side automatic speech recognition (ASR) system and semantically processed to generate a response that is sent back to the device [59]. The processing in the server includes an analysis of the context of the received audio. As explained in [60], the context is “on the location that the user is in, on the time of the day, the user’s search history, the particular dialog state that the user is in, the conversation topic, the content on the screen that the user is looking at, etc.”. This may be another reason that leads to different interpretations of the same input: the variations of the context in a context-dependent interpretation of the user’s speech.

Other issues, such as the type 1, 3, and 6 errors, were related to the way the system interacted with the virtual assistant. The system presented here was based on one-way communication through voice commands from the application running in the laptop to the virtual assistant running in the smartphone. Once a voice command was sent, the BCI system returned to an NC state, taking for granted that the virtual assistant would interpret this command correctly. However, as previously explained, this assumption was not always correct, since there were some cases in which the assistant had trouble interpreting the voice commands and it asked the user to specify, for example, to which contact the message should be sent. In these cases the users did not have the ability to respond beyond the preset commands available, so the task could not be completed.

On the other hand, some of these issues were related to the experimental environment and one could suppose that in a daily use situation these problems would not appear. This is the case of type 2, 3, and 4 errors, whose problems arose because of the need to add new e-mail addresses and contacts with each new subject; this situation is not applicable in daily use, as it is expected that a real user would send messages to already registered contacts.

## 5. Conclusions

This paper has presented a BCI system that allowed the control of four of the most widespread messaging applications in a smartphone (WhatsApp, Telegram, e-mail, and SMS). To our knowledge, this is the first BCI proposal using WhatsApp and SMS. The BCI interaction relies on a speller-like RCP paradigm combining menus with control commands and spelling items. The communication between the BCI part and the messaging applications is achieved through the use of voice commands sent to the virtual assistant present in the smartphone.

The performance results obtained are similar to previously published research on BCI-based spelling systems and the subjective questionnaires regarding usability and workload are fairly positive. Even when this might suggest that the whole system could be useful for severely motor-disabled patients, some aspects should be considered when extending the conclusions from healthy subjects to patients with motor disabilities. When using visual P300 RCP paradigms, the accuracy is generally lower with patients than with healthy subjects, and consequently the number of patients reaching the mentioned threshold of 70% (minimum for efficient communication) is also lower. For example, in [61,62,63] the percentage of patients achieving this threshold was 67%, 68%, and 63%, respectively. These studies did not find a correlation between the degree of impairment (as measured by the ALS Functional Rating Scale, [64]) and the BCI performance. On the other hand, as concluded in [62,63], the difficulty in controlling the device may be due to lack of gaze control, which is a critical ability to control a visual RCP-based paradigm.

As for the time required, it is kept within reasonable limits as explained in Section 4.1; subjects achieved a mean of 3.05 selections/min. Furthermore, the use of a text predictor brought the spelling parts to an OCM of 4.11 characters/min.

The use of a virtual assistant to control the smartphone makes it possible to easily extend the functionality to other applications beyond messaging services, for example, to control domotic devices through the smartphone. However, the use of a virtual assistant also presented some misinterpretation problems that have been detailed in the paper, as well as the possible solutions to them.

The future work based on this study is related to the extension of the functionality to domotic features, the improvement of the one-way communication from the BCI application to the virtual assistant, the use of pictograms instead of text in order to improve the user experience, the option of improving the system by studying new input signals and more sophisticated decoding algorithms, and the possibility of testing the system with motor-disabled patients in real environments.

## Figures and Tables

**Figure 1 sensors-21-03716-f001:**
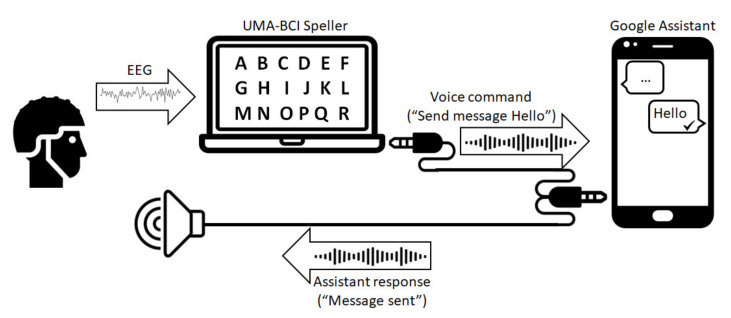
System implementation. By using an oddball paradigm the user can select items from a matrix to spell a command, e.g., “Send message Hello”. This text command is converted to voice in the computer and sent through an audio cable to the microphone input of a smartphone. The smartphone virtual assistant receives the command, performs the corresponding action (to send the message “Hello”, in the example), and gives audio feedback to the user (“Message sent”, in the example) through a loudspeaker.

**Figure 2 sensors-21-03716-f002:**
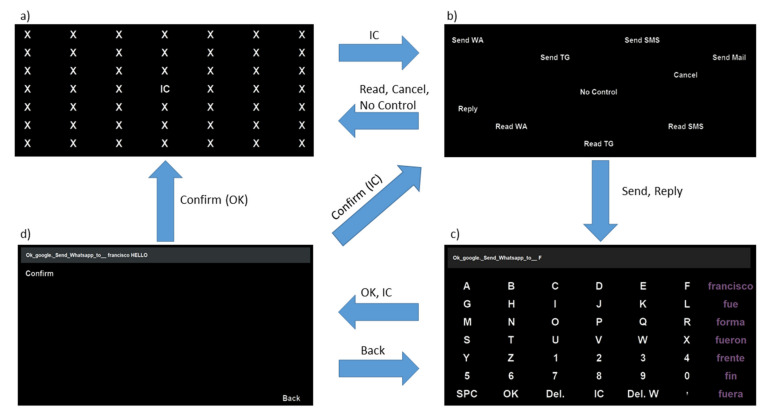
Navigation scheme between the four available menus: (**a**) No Control, (**b**) Intentional Control, (**c**) Spelling, (**d**) Confirmation. The images shown in this figure are the English translations of the original Spanish commands.

**Figure 3 sensors-21-03716-f003:**
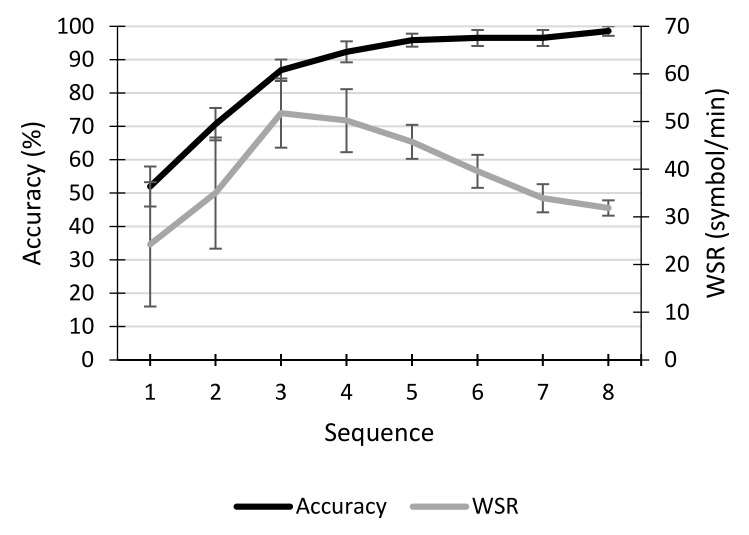
Average (±standard error) accuracy (%) and written symbol rate (WSR; symbol/min) obtained by the participants in each of the sequences in the calibration session.

**Figure 4 sensors-21-03716-f004:**
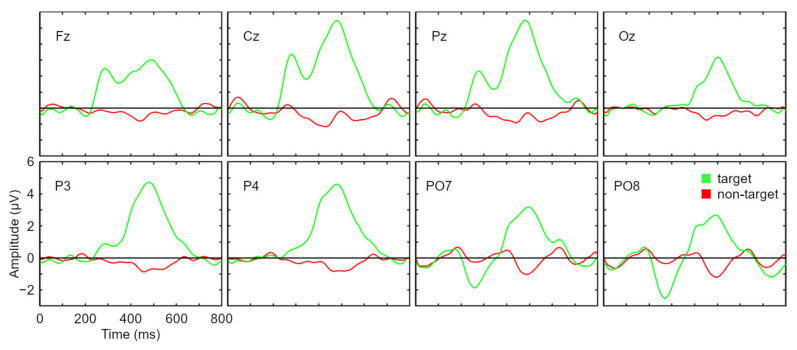
Grand average event-related potential waveforms (µV) for target and non-target stimuli in each electrode position.

**Figure 5 sensors-21-03716-f005:**
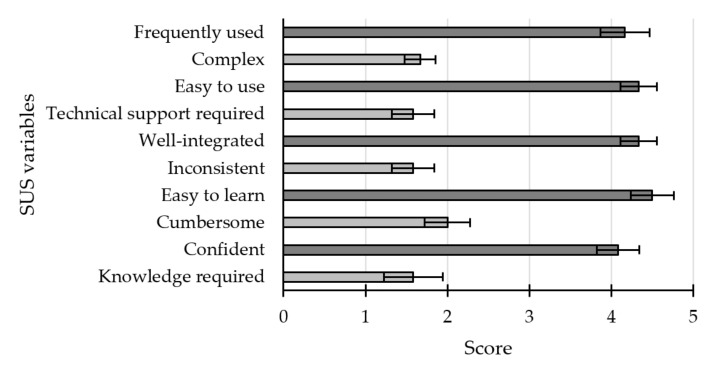
Average results (±standard error) reported by participants in each of the subdimensions of the system usability scale (SUS). The dark grey scales refer to positive items, while the light grey scales refer to negative items.

**Figure 6 sensors-21-03716-f006:**
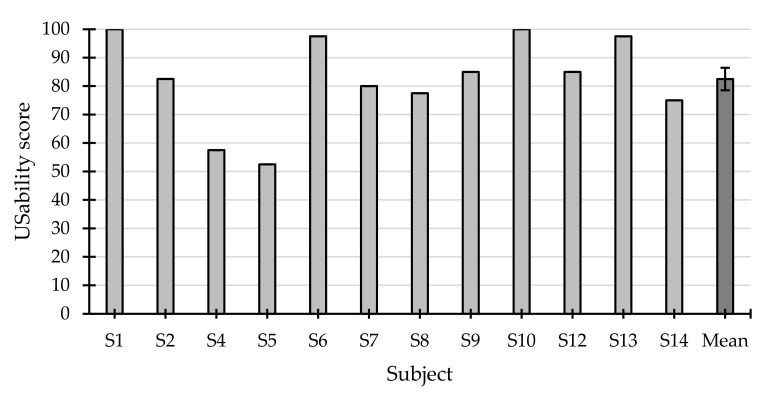
Overall usability score of each subject, measured with the system usability scale, and the average (±standard error).

**Figure 7 sensors-21-03716-f007:**
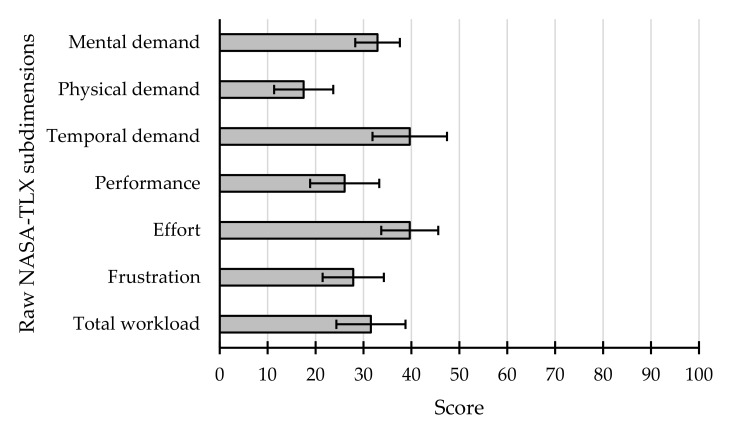
Average results (±standard error) reported by participants in each of the subdimensions of the Raw NASA-TLX.

**Figure 8 sensors-21-03716-f008:**
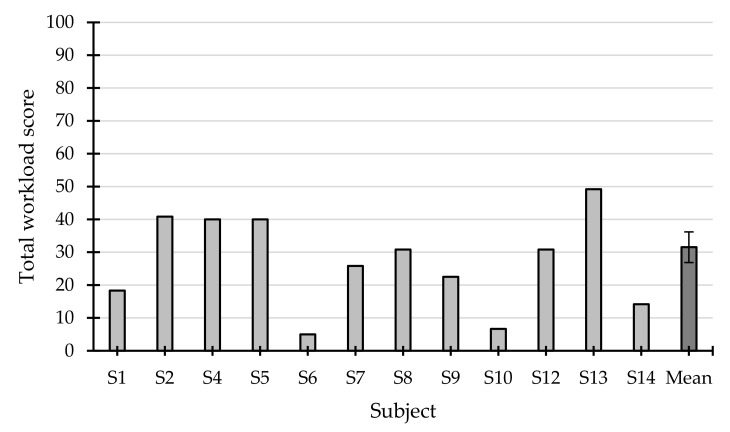
Total workload score of each subject, measured with the Raw NASA-TLX, and the average (±standard error).

**Table 2 sensors-21-03716-t002:** For each task (column), the actions required to complete the task are detailed (right sub-column) as well as the menu in which they should be performed (left sub-column). The normal text corresponds to specific command selection actions (e.g., IC or Send WA); italics correspond to grouped actions (e.g., *Spell receiver*) or to more general instructions (e.g., *Rest*).

Task 1		Task 2		Task 3		Task 4	
Menu	Actions	Menu	Actions	Menu	Actions	Menu	Actions
NC	(*Wait 1 min*)	NC	(*Rest*)	NC	(*Rest*)	NC	(*Rest*)
	- IC		- IC		- IC		- IC
IC	- Send WA	IC	- Read SMS	IC	- Read TG	IC	- Send Mail
Spelling	- *Spell receiver*			NC	- IC	Spelling	- *Spell receiver*
	- *Spell message*			IC	- Reply		- *Spell message*
	- OK			Spelling	- *Spell response*		- OK
Conf.	- Confirm				- OK	Conf.	- Confirm
				Conf.	- Confirm	NC	(*Wait 1 min*)

**Minimum actions**							
19		2		6 + free spelling response		7 ^1^ + free spelling message	

^1^ Each chosen e-mail receiver was included in the application’s prediction corpus so that it was suggested as a prediction when the first two characters were selected. Therefore, three actions were needed to select the receiver (these actions were included in the first seven necessary actions).

**Table 3 sensors-21-03716-t003:** Online task results in terms of number of sequences used (Seq), selections made (Selected commands), time to complete each task (Time required), and overall accuracy (Acc). The sub column Total in Selected commands shows two values: the first one is the sum of Task 1 to Task 4, the second value represents the minimum number of actions needed to complete the four tasks. As previously shown in Table 2, this minimum number of actions varies among subjects as it depends on the free message they chose.

Subject	Seq.	Selected Commands					Time Required (s)					Acc
		Task 1	Task 2	Task 3	Task 4	Total	Task 1	Task 2	Task 3	Task 4	Total	
S1	3	19	2	29	21	71/67	304	32	464	336	1136	98.59
S2	4	31	3	28	31	93/57	591	57	534	591	1773	78.5
S4	4	23	3	27	28	81/62	438	57	514	533	1542	86.42
S5	5	20	3	23	24	70/55	444	67	510	533	1554	88.57
S6	4	23	2	25	22	72/63	441	38	479	422	1380	93.06
S7	8	25	3	24	39	91/59	789	95	757	1231	2872	80.22
S8	3	30	3	28	52	113/65	475	48	444	824	1791	70.8
S9	6	19	2	28	22	71/60	485	51	714	561	1811	92.96
S10	4	19	3	23	39	84/57	362	57	439	744	1602	85.71
S12	3	21	2	21	26	70/61	335	32	335	414	1116	94.29
S13	3	23	2	50	37	112/73	365	48	794	588	1795	75
S14	4	25	2	16	24	67/55	476	38	305	457	1276	89.55
Mean	4.25	23.17	2.5	26.83	30.42	82.92/61.17	458.8	51.67	524.08	602.8	1637.33	86.14
Standard deviation	1.49	4.06	0.52	8.19	9.51	16.27/5.3	130.1	17.51	155.69	240.5	463.06	8.46

**Table 4 sensors-21-03716-t004:** Issues with the virtual assistant.

Subject	Task	Description 3	Error Type
Preliminary tests, S13	3	Time limit to reply to an incoming message	1
S2	4	Use of the same name for different contacts	2
S4	4	Use of a similar name for different contacts	3
S7	4	Use of an unrecognisable word for a contact	4
S8	3	Phonetic similarity	5
S8	4	Confused subject/body of the message	6
S11 ^1^	3	Automatic correction of misspelled words	7
S14	4	Words missing	8


^1^ This participant was discarded from the results section, see Section 2.1 for details.

## Data Availability

The data presented in this study are available on request from the corresponding author. The data are not publicly available due to data privacy restrictions.

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
