# Peer review of "Brain–Computer Interface (BCI) Control of a Virtual Assistant in a Smartphone to Manage Messaging Applications"

_sensors, 2021, doi:10.3390/s21113716_

Round 1
Reviewer 1 Report
The paper describes a BCI system to control 4 messaging applications in a smartphone throught a Google virtual assistant. The application can be of uttermost importance for patients with illnesses like ELA that progressively loose their ability of communication with their social environment. The pilot study was done with 13 non-disabled students of the University and consisted on four tasks of sending and receiving mesages through whatsapp, telegram, SMS and mail. The study analyzed the performance, times and opinions of users through different questionnaires (SUS, raw NASA TLX, ad-hoc questions). The methods and well described and analyzed and the sources of error are discussed. The basic concern about the paper (explaining the "average" assessment) is that the study is very limited because the number of users is low and none of them have any kind of disability. The results can be completely different with subjects having the targeted illnesses. Thus, jumping to the conclusions that “the whole system could be useful for severely motor-disabled patients” on the basis of the performance results and subjective questionnaires is too daring. The discussion misses an explanation based on clinical experience of what limitations real patients can present. For instance, what can be the expected accuracy? Are times acceptable?
For future works, authors could consider the use of illustrations or icons that can be a faster way to convey emotions than text.
Reviewer 2 Report
The authors proposed to integrate UMA-BCI and Voice Assistant for further application. These techs are not new, but the integration may bring something potential, especially for the patients. The authors well constructed the whole manuscript with clear descriptions. Here are some points as reading the paper.
- The introduction clearly lists multiple BCIs. Obviously, there are some BCIs (such as SSVEP or ASSR) are more reliable as well as stable. Can the authors provide more reasons and information to explain why P300 and ERP are the main BCI in this study?
- Regarding ERP or P300 analysis/classification, the SWLDA was applied. Can the authors compare the more approaches for comparison purposes? There are some new tools in ML, DL and AI domains may significantly improve the performance of identifying P300.
- Most information about EEG signals processing and feature extraction may be missing. It would be very important to include the information for further validating classification performance. Meanwhile, how did the authors fight to subject difference? The authors only listed the individual parameters for classification. How and What to decide the individual parameters would be important as well.
- Moreover, how did the authors process the noise during recording as well as EEG signal processing?
- I am wondering to know the relationship between the performance and the number/order of sessions, especially the online sesssion. According to the previous study, time-of-task or fatigue would be one important factor to BCI performance. As considering the whole procedure (calibration and online sessions), this fatigue would be worth considering.
a. M Nascimben et al., Alpha correlates of practice during mental preparation for motor imagery, IEEE TCDS 2020
b. B J Lance et al., Towards serious games for improved BCI, 2016. - The questionnaires are pretty great to include in the BCI study. For comparisons, the authors also need to consider the statistic tests.
- The EEG phenomenon would be worth showing.
Reviewer 3 Report
In the considered manuscript, the authors build and evaluate an interface system between a human brain and messaging apps on a smartphone. In my opinion, the paper addresses a relevant topic, has a solid methodological basis and good technical quality, and is generally well written. I would recommend accepting it for Sensors, provided that some concerns that I do have are addressed (see below).
Most importantly, I believe the authors need to better justify feasibility of their solution's architecture. Particularly, what are the benefits of using the voice commands and the virtual assistant? Clearly, this link in the information transmission is problematic, which the authors duly note in the sub-section 3.3. It is also outside of the authors' and the potential users' control, as it relies on a third-party cloud solution that is also AI-based and stochastic. Again, the authors do note the disadvantages and risks, but I feel that they don't provide enough arguments for the voice transmission link. Given that it is a crucial element in their solution and presumably the key aspect of the work's originality, this must be fixed.
Also, I would like to see comparisons to alternatives - e.g. using just a data interface in this link. EEG registration equipment is pretty sophisticated already, so even coding the interface for the solution might be justified.
Second, I believe the authors should discuss the generalizability of their experimental findings. The participants they employed were healthy, but can they be representative of the potential target users of the solutions, who have neurological problems? Their EEG patterns and the over user experience might be different. Commonly, user testing sessions are performed with representative users, so the deviation should be explained.
Finally, there are some minor limitations that in my option also deserve more detailed discussion of their impact:
- 2 participants were dropped due to low accuracy - how realistic would such problems be in a real environment?
- "using a single-word name for each contact, the user’s given name." - this seems to be not common in reality?
- the authors mention "Spanish language specific corpus" - so does the solution generalize to all other languages?
- usability scores of 100% (Fig. 5) - this seems somehow unrealistic, did the participants have any positive bias towards the solution?
Round 2
Reviewer 2 Report
The authors have a huge effort to the current version. Here are 2 more points.
- Regarding EEG signal processing, having the key steps would be very important and helpful to the readers. At least the reader can have a rough concept of the processing steps as reading the work. If the readers would like to understand more, they can read the citations. The current version would be sufficient to meet the authors' claim.
- The authors explained that the results of questionnaires were from part of the participants. However, I can see the individual scores through the results. My idea is to compare the results of questionnaires at different stages of the experiment. For example, the fatigue factor, learning effect, and workload can be assessed clearly.
